# Monocarboxylate Transporters Are Involved in Extracellular Matrix Remodelling in Pancreatic Ductal Adenocarcinoma

**DOI:** 10.3390/cancers14051298

**Published:** 2022-03-02

**Authors:** Ayşe Ufuk, Terence Garner, Adam Stevens, Ayşe Latif

**Affiliations:** 1Division of Pharmacy and Optometry, Faculty of Biology, Medicine, and Health, University of Manchester, Manchester M13 9PT, UK; ayshe.ufuk@manchester.ac.uk; 2Division of Developmental Biology and Medicine, Faculty of Biology, Medicine, and Health, School of Medical Sciences, University of Manchester, Manchester M13 9WL, UK; terence.garner@manchester.ac.uk (T.G.); adam.stevens@manchester.ac.uk (A.S.)

**Keywords:** pancreatic cancer, transcriptomics, hypernetworks, tumour microenvironment

## Abstract

**Simple Summary:**

Monocarboxylate transporters (MCTs) carry a variety of substrates, with MCT1–4 being well-characterised and involved in the proton-coupled transport of monocarboxylates (such as lactate), which can be used as metabolic fuel for cancer cells. Increased acidity of the tumour microenvironment via MCTs favours remodelling of the extracellular matrix (ECM), leading to desmoplasia associated with tumour metastasis and poor patient outcomes. Although MCT1, MCT2, and MCT4 are upregulated in several cancers, their expression and role in pancreatic ductal adenocarcinoma desmoplasia is little understood. Here, we aimed to understand the role of MCTs in desmoplasia through their association with ECM components. Our analysis using hypernetworks showed the presence of bidirectional associations of MCTs and ECMs, suggesting the presence of a causal relationship and the need to further investigate their functional associations. It confirmed the role of MCTs in desmoplasia, highlighting their importance as therapeutic targets alone or in combination with key ECM components to potentially improve patient outcomes.

**Abstract:**

Pancreatic ductal adenocarcinoma (PDAC) is an aggressive malignancy with a five-year survival rate of <8%. PDAC is characterised by desmoplasia with an abundant extracellular matrix (ECM) rendering current therapies ineffective. Monocarboxylate transporters (MCTs) are key regulators of cellular metabolism and are upregulated in different cancers; however, their role in PDAC desmoplasia is little understood. Here, we investigated MCT and ECM gene expression in primary PDAC patient biopsies using RNA-sequencing data obtained from Gene Expression Omnibus. We generated a hypernetwork model from these data to investigate whether a causal relationship exists between MCTs and ECMs. Our analysis of stromal and epithelial tissues (*n* = 189) revealed nine differentially expressed MCTs, including the upregulation of *SLC16A2/6/10* and the non-coding *SLC16A1-AS1*, and 502 ECMs, including collagens, laminins, and ECM remodelling enzymes (false discovery rate < 0.05). A causal hypernetwork analysis demonstrated a bidirectional relationship between MCTs and ECMs; four MCT and 255 ECM-related transcripts correlated with 90% of the differentially expressed ECMs (*n* = 376) and MCTs (*n* = 7), respectively. The hypernetwork model was robust, established by iterated sampling, direct path analysis, validation by an independent dataset, and random forests. This transcriptomic analysis highlights the role of MCTs in PDAC desmoplasia via associations with ECMs, opening novel treatment pathways to improve patient survival.

## 1. Introduction

Pancreatic cancer is the 12th most common cancer and the seventh-highest cause of cancer mortality worldwide [1]. In the UK, it is the 10th most common malignancy, with its incidence rates predicted to increase 6% by 2035 [2]. PDAC is the most common malignancy of the pancreas, accounting for 95% of all pancreatic cancer cases, with a five-year survival rate of less than 8% [3,4]. The poor clinical prognosis of PDAC stems from the difficulty of its diagnosis, ineffective treatment options at advanced stages, and surgery being the only existing curative option that can be offered to only 10–20% of patients with resectable PDAC [4,5,6]. The most well-known mutations in PDAC include *KRAS*, *TP53*, *CDKN2A*, and *SMAD4*, with *KRAS* being present in more than 90% of patients with PDAC, and are considered as initiators of tumorigenic progression [7]. In pancreatic cancer, K-Ras signalling activates downstream effector pathways, including Raf/MAPK, PI3K/Akt, Ral-GEFs, and phospholipase Cε, which have important roles in cell cycle regulation, apoptosis, cell growth, differentiation, and migration, as well as the regulation of various transcription factors [8,9,10,11,12]. Progression from the earliest pancreatic intraepithelial neoplasias to invasive PDAC is accompanied by the inactivation of *CDKN2A* as the most frequently seen tumour suppressor gene in pancreatic cancer, followed by the loss of function mutations in *TP53* and *SMAD4* [7,13].

A key feature of PDAC is the existence of extensive fibrotic stroma, also known as desmoplasia, which has an abundance of dense extracellular matrix (ECM) surrounding fibroblasts, immune cells, endothelial cells, and neuronal cells [14,15]. The components of ECM include collagens; glycoproteins (e.g., laminins, elastin, and fibronectin); and proteoglycans (e.g., hyaluronic acid), as well as ECM remodelling enzymes involved in degradation or cross-linking of the ECM components (e.g., matrix metalloproteinases (MMPs) and their tissue inhibitors (TIMPs) and lysyl oxidases (LOXs)) [16,17,18]. The excessive accumulation of ECM components has been highlighted as a major contributor to PDAC progression and the resistance to therapeutic efficacy [5,16]. ECM remodelling occurs as a result of the crosstalk between tumour cells and the microenvironment, including pancreatic satellite cells (PSCs) in stroma, which cause aberrant secretion of ECM components by the secretion of profibrotic and inflammatory growth factors.

Monocarboxylate transporters (MCTs) are key players in cellular metabolism and have an important role in regulating intracellular pH [19]. They belong to the solute carrier 16 (SLC16) gene family, with MCT1/*SLC16A1*, MCT2/*SLC16A7*, MCT3/*SLC16A8*, and MCT4/*SLC16A3* being well-characterised in terms of their role in the proton-coupled transport of monocarboxylates such as lactate, pyruvate, and ketone bodies. From a metabolic standpoint, most solid tumours rely on glycolysis to produce energy, leading to the production of large amounts of lactate that is exported by the MCTs [20,21]. Therefore, MCTs help cancer cells maintain their high glycolytic activities and contribute to tumour acidosis and progression [22,23]. MCT1, MCT2, and MCT4 are known to be upregulated in different cancers, and a high expression of MCT1 and MCT4 is often associated with a poor prognosis [20,22]. MCTs contribute to the acidification of the tumour microenvironment, which is known to activate and increase the expression of the ECM remodelling enzymes involved in ECM degradation, contributing to the desmoplastic reaction [24,25]. However, their expression and association with desmoplasia are still largely unknown in PDAC.

So far, the association between MCTs and ECM components has been shown in a limited number of cancer types in vitro. For example, the upregulation of MCT1 was associated with increased invasiveness and migration of nasopharyngeal carcinoma cell lines accompanied with increased expression of MMP2 and MMP9 and downregulation of TIMP1 and TIMP2 [26]. Secondly, overexpression of lysyl oxidase-like 1 (LOXL-1) in a non-small cell lung cancer metastasis model was associated with the upregulation of MCT1/2, increased expression and activity of MMP2/9, and increased cell migration and invasion, along with increased extracellular lactate at a low pH [27]. The latter findings also support the correlation of acidic pH via lactate transport by the MCTs with increased activity of the MMPs [18,21,22]. However, in PDAC, the link between MCT and ECM components has not been elucidated. Therefore, it is vital to investigate the correlation of MCTs with ECMs and consolidate the role of MCTs in PDAC desmoplasia, as well as bring forward novel treatment strategies co-targeting MCTs and key ECM players to reduce tumour progression and improve patient survival.

Omics approaches are now widely used in the field of cancer to identify dysregulated molecular mechanisms, provide insights into biological pathways, and propose potential biomarkers for different cancer types. The incorporation of higher-order interactions driven from omic data is now seen as an essential part of modelling biological systems [28]. Hypernetworks model higher-order interactions or relationships between omic elements (represented by nodes) based on large numbers of shared correlations (represented by edges) [29]. Such interactions that are normally not captured by traditional pairwise transcriptomic approaches provide a model of functional relationships between these omic elements [30,31].

In this study, we set out to evaluate MCT–ECM interactions using in silico approaches utilising data available in a public repository, the National for Biotechnology Information (NCBI) Gene Expression Omnibus (GEO). To achieve this, we investigated the causal link between MCT and ECM interactions using a hypernetwork model of PDAC transcriptome based on literature-collated RNA-sequencing data. Our hypothesis and study design related to the causal analysis is illustrated in Figure 1. The purpose of the hypernetworks is to decipher the high-dimensional relationships between multiple transcripts by flattening them into a single metric of similarity using the summary of the correlation matrices [29]. In order to refine the hypernetwork model, the differential expression of MCT and ECM genes were first determined. The findings of the causal analysis were validated using independent approaches. In addition, we sought to understand the correlation between MCT and ECM expression with age at PDAC diagnosis. Finally, we assessed MCT and ECM expression in short- and long-term survivors of patients with PDAC.

## 2. Materials and Methods

### 2.1. Study Selection and Data Processing

The NCBI GEO repository browser (https://www.ncbi.nlm.nih.gov/geo/browse/, accessed on 15 December 2021) was used to search for RNA-sequencing (RNA-seq) studies of pancreatic cancer involving “pancreatic” as a search term, “*Homo sapiens*” as an organism, and “expression profiling by high-throughput sequencing” as a series type. Studies with RNA-seq data from primary PDAC tissue biopsies with their matching controls where available were included in the analysis.

The quality control for the raw sequencing data was performed with FastQC v0.11.9 (https://www.bioinformatics.babraham.ac.uk/projects/fastqc/, accessed on 19 December 2021). The raw sequencing reads were filtered using BBDuk from the BBMap toolkit v38.87 (ktrim = r, k = 21, mink = 9, hdist = 1; https://sourceforge.net/projects/bbmap/, accessed on 19 December 2021) to remove adapters and quality trim both ends to Q15 (qtrim = rl, trimq = 15, minlength = 36). Read mapping to human reference genome hg38 with Gencode v35 annotations and gene quantification using “--quantMode GeneCounts” were performed with STAR v2.7.6a [32]. Post-mapping quality control was performed with RseQC v4.0.0 (http://rseqc.sourceforge.net/, accessed on 19 December 2021).

### 2.2. Segregation Analysis

To evaluate the quality of the collated transcriptomic data and assess the similarities of different datasets, we performed the unsupervised learning method Principal Component Analysis (PCA). For the analysis of all collated studies (*n* = 4), each study was initially processed independently to remove genes with low expression (genes with fewer than 10 reads) using the R package edgeR (version 3.32.1 [33]). Once genes with low expression were filtered out, the raw count table, including all datasets, was trimmed down to these common genes. Trimmed Mean of M-values (TMM) normalisation was used to facilitate a comparison of expressions between samples. These normalised data were then used to conduct PCA using the R package mixOmics (version 6.14.1 [34]). The analysis was run with the pca function based on the calculation of the first two principal components while both centring and scaling of the data were applied.

Segregation of the stromal and epithelial dataset alone (GSE93326) based on tumour overall stage, grade, and N-score was also performed by applying each of these features as a factor in the analysis.

### 2.3. Differential Gene Expression Analysis

Differentially expressed MCT, ECM, and ECM-related genes were identified using the R package edgeR. When analysing differentially expressed genes (DEGs) in the stromal and epithelial dataset alone (GSE93326), the comparison was performed between the stroma and epithelium, which included 66 matched tissues and an additional 57 stromal data in comparison to the analysis conducted by Maurer et al. (2019) (analysed 60 matched tissues) [35]. The analysis of DEGs in short- (STS) and long-term survivors (LTS) from the GSE79668 dataset was performed using tumour tissues from the mentioned survival groups (14 and 13 samples from STS and LTS, respectively). When analysing DEGs between tumour and non-tumour tissues from all datasets combined (*n* = 4), the pre-processing of the studies in terms of the removal of low expression genes and data normalisation was handled as described above. The gene expression levels were calculated as log2 count per million (CPM). The difference in the gene expression levels was calculated as the log2-fold changes (logFC) of genes between the comparison groups. Genes were filtered using a false discovery rate (FDR)-adjusted *p*-value < 0.05 to determine the significance. Hierarchical clustering analysis of the DEGs was done by the complete method with Euclidean distance. The gene expression profiles were visualised with heatmaps using a modified version of the heatmap.2 function of the R package gplots (version 3.1.1) [36] to allow simultaneous visualisation of multiple annotations. Log-normalised expression levels of the MCT genes were visualised with violin plots using the R package ggplot2 (version 3.3.5) [37].

We identified differentially expressed ECM and ECM-related genes based on the NABA_MATRISOME gene set (version 5.0) from the Molecular Signature Database (MSigDB) as a reference [17]. This gene set is an ensemble of 1026 genes that encode ECM and ECM-associated proteins. Due to their roles in cell adhesion and migration and potential contributions to ECM remodelling and stiffness [11,33,34], we also included integrins and keratins into the matrisome gene set to provide a broader coverage of the ECM-related genes.

The conversion of Ensembl IDs to gene names was performed either by using the R package biomaRt (version 2.46.2) [38] or the BioMart online tool from Ensembl (Ensembl release 104) [39].

### 2.4. Causal Analysis

The causal relationship between MCTs and ECMs was evaluated using a hypernetwork modelling approach as previously described [29,31]. Briefly, hypernetworks represent network structures where edges define a relationship between nodes (e.g., transcripts) and can be shared by many nodes (Appendix B, Figure A1); this is the definition of “higher order interactions” [28]. In this way, hypernetwork structures are used to delineate complex relationships that connect multiple omic elements. We can determine the connectivity of target elements (e.g., differentially expressed genes) within the whole transcriptome by quantifying their number of shared edges (i.e., correlations or gene–gene interactions). This not only provides a summary of higher-order interactions from correlation matrices but also implies functional relationships between strongly associated elements [29].

Here, we used hypernetworks to evaluate the correlations between the MCT and ECM (and ECM-related) transcripts within the stromal–epithelial dataset (GSE93326). The differentially expressed MCTs and ECMs were used to refine the selection of target genes for the hypernetworks. To assess the similarity of the differentially expressed MCTs (*n* = 9) or ECM and ECM-related (*n* = 502) transcripts, we first determined the Pearson correlation coefficients as a distance metric between each of these transcripts and the rest of the transcriptome (*n* = 13,815 and *n* = 13,322 in the case of correlations with MCTs and ECMs, respectively). Values were binarized using ±1 standard deviation (SD) from the mean, such that the direction of the correlations (i.e., negative or positive) was ignored. This formed the matrix *M*, where
 x>|1|sd, then x=1 x≤|1|sd, then x=0 

The *sd* of the r-values were 0.20 and 0.28 for the MCT and ECMs, respectively, with their distribution profiles shown in Figure A2. The binarized matrix *M* is the incidence matrix of the hypernetwork where target genes are nodes and non-target genes are edges. This matrix was then multiplied by its transpose *M^t^* to generate a new matrix ($M\times M^{t}$), which describes the number of shared correlations between any pair of DEGs. This final matrix represents the adjacency matrix of the hypernetwork, which describes the higher-order interactions between the genes and has been suggested to describe functional relationships [30]. Hierarchical clustering was used to identify the clusters formed within the hypernetwork. The correlations between the highly connected cluster of transcripts from the hypernetwork and the rest of the transcriptome were determined by interrogating the incidence matrix, *M* [31]. This produced a subset of the whole transcriptome that correlated with 90% of the transcripts from the hypernetwork cluster.

The robustness of the hypernetworks was evaluated using a number of approaches. Firstly, we tested if the interactions between MCT and ECM genes observed in the hypernetwork model were more frequent than expected by random chance. This was achieved by iterating one thousand hypernetworks using 7 and 376 randomly selected genes for MCTs and ECMs, respectively, based on the findings of the primary hypernetwork analysis. The mean number and SD of ECM and MCT genes that were found to correlate with these random genes were calculated.

Secondly, we refined a network of MCTs and ECMs by silencing the indirect relationships between genes, retaining only the direct links [40]. To demonstrate the presence of direct links between the MCTs and ECMs, we first correlated differentially expressed MCT and ECM genes (9 and 502, respectively) together. Indirect relationships in this correlation matrix were silenced using a modified version of Equation (5) of Barzel and Barbasi (2013) [40], which incorporated the Moore–Penrose approximation of the inverse of the correlation matrix. Consequently, matrix *S* represents the directness of the relationship between gene pairs. The interaction directness was ranked to identify the strongest interactions between connected nodes. The significance of these interactions was calculated as a z-score (relationship directness score-mean(S))/sd(S)) and the associated *p*-values calculated from the cumulative distribution function of the normal distribution.

To provide further validation for the causal relationship between MCTs and ECMs, we used an independent RNA-seq dataset (GSE164665) from Birnbaum et al. (2021) including 19 pairs of matched epithelium and stroma samples. The raw count data were normalised and scaled as described in the previous section. Hypernetwork models of these data were generated in the same manner as our primary dataset (GSE93326): the differentially expressed MCT and ECM genes that clustered in the primary dataset were used to form the causal model, and the analysis was conducted bidirectionally (i.e., from MCT to ECM transcripts and vice versa). Using the hypernetwork incidence matrix, the percentage of target genes from the central cluster that interact with nontarget ECM and MCT genes was determined. A Wilcoxon rank-sum test on these proportions was used to test whether the ECM and MCT genes interacted more with one another than with any other genes.

Following the generation of the hypernetwork models, a direct path analysis was performed as described above; the differentially expressed MCT and ECM genes from the primary dataset (9 and 502, respectively) were correlated based on the expression values from the validation (Birnbaum) dataset. The strength of the interactions found between ECM and MCT genes in the primary dataset was assessed in the validation dataset.

Finally, a random forest (RF) model was generated using the primary dataset to predict the tissue phenotype from the gene expression data. We used a Synthetic Minority Oversampling TEchnique (SMOTE, R package smotefamily) [41] to balance the classes before partitioning the dataset into training (70%) and test (30%) sets. In addition, we used the Birnbaum data as validation for these models. RF models were generated using two sets of genes: (1) a combination of the MCTs and ECMs identified by the hypernetwork model and (2) a representative-sized set of non-differentially expressed genes in the primary dataset. The statistical parameters used to evaluate the model performance were the model out-of-bag (OOB) error rate, area under the curve (AUC) with 95% confidence interval (CI), and AUC of the validation set and error rates for the test and validation sets. These parameters were estimated using the R packages pROC, ROCR, caret, and randomForest, respectively [42,43,44,45].

All primary analyses were performed in R (version 4.0.3). The silenced matrix resulting from the direct path analysis was visualised in Cytoscape (version 3.9.0) and imported using the aMatReader app (version 1.2.0).

### 2.5. Functional Annotation

Functional annotation of the genes that showed a correlation with the clustered MCT and ECM genes in the hypernetwork analysis was conducted using the online tool DAVID [46,47].

### 2.6. Investigation of the Correlation between MCT–ECM Expression in Stroma and Epithelium with Age at PDAC Diagnosis

To investigate how the levels of differentially expressed MCT, ECM, and ECM-related genes in stromal and epithelial tissue samples associate with age at diagnosis of PDAC, we performed Pearson correlation analysis using the rcorr function of the R package Hmisc (version 4.5-0; Hmisc: Harrell Miscellaneous (uib.no), accessed on 19 December 2021). Log-normalised TMM data for 187 samples from the GSE93326 dataset with available age information were used in the correlations. Both the correlation coefficients and the *p*-values were reported.

### 2.7. Assessment of MCT, ECM, and ECM-Related Gene Expression in Short- and Long-Term Survivors

We evaluated the levels of MCT, ECM, and ECM-related genes identified as differentially expressed in stromal-epithelial dataset in short- and long-term survivors from the GSE79668 dataset (14 and 13 subjects, respectively). Violin plots were used to inspect the log-normalised expression levels of the top four MCT genes that showed differential expression (*p* < 0.05) in the stromal–epithelial samples. ECM and ECM-related genes that were significantly different (FDR < 0.05) in both the stromal-epithelial and STS–LTS datasets were examined with a heatmap using the modified heatmap.2 function as described previously.

## 3. Results

### 3.1. Large Interstudy Differences in Gene Expression Levels Exist in PDAC

In this study, we aimed to establish a link between MCTs and ECMs using publicly available transcriptomic data. Based on the criteria outlined (RNA-seq, primary PDAC tissue biopsies, and *Homo sapiens*), four RNA-seq datasets were included in the initial analysis, as summarised in Appendix A [35,48,49,50]. In addition to the studies reporting RNA-seq data from primary PDAC bulk tissues, a study with GEO accession number GSE93326 with data from PDAC stroma and epithelium was selected to help delineate the role of MCTs in desmoplasia. One study was excluded from the analysis due to RNA-seq data being generated from cell lines that were isolated from primary tumours (GSE63124).

The PCA revealed large variations in the transcriptomic data from tumour tissues between different datasets, whereas minimal differences were observed between tumour and non-tumour tissues (Figure A3). Although the differences between tumour tissues and studies were not clear in the PCA plots when all samples were included in the segregation analysis (Figure A3a,b), this difference was highlighted when tumour stromal and epithelial data were excluded, and the GSE79668 dataset was compared with the remaining datasets (Figure A3c,d). As a result, we deemed the collated RNA-seq datasets on tumour and non-tumour tissue biopsies inappropriate for the purpose of conducting a differential gene expression analysis and focused on the analysis of epithelial and stromal data from the GSE93326 dataset.

### 3.2. MCT, ECM, and ECM-Related Genes Are Differentially Expressed in PDAC Stroma and Epithelium

The PCA of the GSE93326 dataset (including data from 66 paired epithelium and stroma and additional 57 stroma) showed two distinct groups of epithelial and stromal samples as similar to those previously observed by Maurer et al. (2019) for the 60 paired samples [35]. We further investigated whether the separation of epithelial and stromal data was influenced by cancer severity by segregating the samples based on the established PDAC covariates (which were made available in the GEO database) [51,52,53]: overall tumour stage, grade, and metastasis to nearby lymph nodes. The distinct separation of the stroma and epithelium was not explained by any of the individual factors (i.e., tumour stage), as their occurrence was homogenous among the samples (Figure A4).

Next, we conducted a differential gene expression analysis to determine the MCT and ECM genes associated with epithelium and stroma. The analysis showed 13,824 features in 189 samples (123 stroma and 66 epithelium) following the filtering of low-expression genes. Of these features, 8944 DEGs were identified between the PDAC stroma and epithelium samples (FDR < 0.05), including 4481 upregulated and 4463 downregulated genes in the stromal compartment (Figure A5 and Appendix A). The genes found to be upregulated in both compartments showed agreement with those found by Maurer et al., (2019) [35]. The transcriptional profiles for PDAC stroma were well-distinguished from the epithelial counterparts, consistent with the trend observed in PCA (Figure A6).

Nine MCT genes were identified as differentially expressed in PDAC stroma (Table 1, FDR < 0.05). Among the MCT genes, *SLC16A10*/MCT10 was the most significant DEG, showing nearly 3.5-fold higher expression in the stroma relative to the epithelium (Figure 2). In addition to the protein-coding *SLC16A* genes, we identified a long non-coding RNA (lncRNA) *SLC16A1-AS1* to be differentially expressed and highly upregulated (logFC of 2.25) in the stroma samples. Two other *SLC16A* genes that showed a higher expression in PDAC stroma were *SLC16A2*/MCT8 and *SLC16A6*/MCT7, with the remaining MCT genes having higher expression levels in the epithelium (Figure 2).

To identify differentially expressed ECM and ECM-related genes, we used the NABA_MATRISOME gene set from MSigDB. Using this gene set as a reference, and by including integrins and keratins, we identified a total of 502 genes (Appendix A), including several collagens, laminins, and ECM remodelling enzymes such *LOX*s and *MMP*s.

### 3.3. There Is a Causal Relationship between MCT and ECM Gene Expression

Once the differentially expressed MCT, ECM, and ECM-related genes in the stroma and epithelium samples were identified, a hypernetwork analysis was conducted to assess the relationship between the MCT (*n* = 9) and ECM (*n* = 502) transcripts by revealing the presence of multiple correlations between the groups of genes.

The hypernetwork analysis identified two clusters of MCT transcripts. The first cluster included seven transcripts (*SLC16A1-AS1*, *SLC16A6*, *SLC16A5*, *SLC16A10*, *SLC16A2*, *SLC16A13*, and *SLC16A7*), which shared >1500 correlations with the rest of the transcriptome, suggesting similarity and strong functional associations between these sets of MCT mRNAs (Figure 3). The second cluster had two MCT transcripts, including *SLC16A4* and *SLC16A9*, that share < 1000 correlations with the rest of the transcriptome, therefore having weaker functional associations with one another and the remaining MCTs.

For the hypernetwork drawn for ECM and ECM-related transcripts, again, two clusters were identified, including a cluster of 376 transcripts sharing approximately 1000–5000 correlations and a second cluster of 126 transcripts sharing relatively fewer correlations with the rest of the transcriptome (Figure A7).

Once we identified the functionally associated MCT and ECM transcripts that showed high connectivity with the rest of the transcriptome, we extracted the transcripts from the hypernetwork incidence matrix, which showed a strong correlation with each set of transcripts. We found 1714 and 2790 transcripts, which correlated with 90% of the seven MCT and 376 ECM transcripts, respectively. Among the transcripts associated with MCTs, 255 of them were ECM-related, including several collagens (*COL1A1/2*, *COL3A1*, *COL4A1*, and *COL5A1*); laminins (*LAMA2/3/4*, *LAMB2*, and *LAMC1/2*), and fibronectin 1 (*FN1*), and the ECM remodelling enzymes lysyl oxidase (*LOX*), lysyl oxidase-like 1/2/3 (*LOXL1/2/3*), and matrix metalloproteinases (*MMP2/7/11/16/19*) (Table 2 and Appendix A). Likewise, when the hypernetwork analysis was run for ECMs, four MCTs were found to associate with them (Table 2). Of the four MCTs that associated with ECM and their related transcripts, *SLC16A2*/MCT8, *SLC16A10*/MCT10, and *SLC16A1-AS1* were differentially expressed in stromal–epithelial samples. Likewise, nearly all ECM and ECM-related genes that correlated with MCTs showed differential expression (*n* = 254, Appendix A). By examining the link between MCTs and ECMs from both directions (i.e., from MCT to ECM transcripts and vice versa) and establishing the bidirectionality of this relationship, we provide evidence for a causal link existing between these sets of genes.

To confirm that the number of observed interactions between MCTs and ECMs was higher than expected by chance, hypernetworks were iterated using random transcripts. In this way, the number of MCTs or ECMs associated with a random selection of transcripts was calculated. The association was defined as previously, where transcripts must be correlated with 90% of the target transcripts. When sampling 376 random transcripts representative of the ECMs (Table 2), we found, on average, 2.45 ± 1.58 MCTs that were associated with 90% of the random transcripts after 1000 iterations. When sampling seven random transcripts, representative of the MCTs, we could not achieve our target of 1000 iterations, as we consistently found no transcripts to be correlated with 90% of our seven random transcripts (Table 3). This demonstrated that MCTs were more closely associated with ECMs than the random transcripts to such an extent that unrelated transcripts could not replicate our methodology. With this approach, the bidirectional causal link between MCTs and ECMs was validated and demonstrated as independent of random chance; however, we also used an alternative approach to verify our results.

An analysis of the direct paths in a correlation matrix containing differentially expressed MCT and ECM genes resulted in a silenced network *S* consisting of 511 nodes (genes). Having ranked directness between node pairs, we found that some interactions between the ECMs and MCTs were among the strongest interactions in the network. Included in this selection were *SLC16A2*/MCT8 and, most commonly, *SLC16A10*/MCT10, which were both among the most highly connected MCT genes in our hypernetwork (Table 4). Among the genes interacting with *SLC16A10*/MCT10 were *HYAL1* and *ANXA10* (*p*-values < 0.01) and *COL4A4* (*p*-values < 0.05). This approach provides further validation, independent of the hypernetworks, for a causal link between MCTs and ECMs.

Additional validation of the causal relationship between the MCTs and ECMs was established by conducting hypernetwork and direct path analyses using the Birnbaum dataset as an independent study. Firstly, examination of the expression patterns in the Birnbaum dataset of the MCTs was highlighted by the primary analysis, which showed a consistent direction of fold change between tissues (Figure A8).

The hypernetwork model of MCTs in this validation dataset highlighted four MCTs: *SLC16A1-AS1*, *SLC16A10*, *SLC16A7*, and *SLC16A5* (Figure A9), which were all among the clustered MCTs in the primary dataset. Interrogation of the incidence matrix of the hypernetwork revealed 111 ECM genes interacting with all four of these MCTs (Appendix A). Seventy seven percent (85) of these ECMs were shown to be consistent with the incidence matrix of the primary data. Furthermore, 88% (98) of these 111 ECMs were also found within the central cluster of the hypernetwork from the primary data. A Wilcoxon rank-sum test showed a significant difference in the strength of interaction between the MCTs and ECMs compared to MCTs and non-ECM genes (*p*-value < 2.2 × 10^−16^).

Generation of the ECM hypernetwork highlighted a central cluster of 275 ECM genes. The hypernetwork incidence matrix revealed *SLC16A10*/MCT10 and *SLC16A1-AS1* as the strongest interacting MCTs, bearing associations with 83% of the ECMs in the cluster, followed by *SLC16A6*/MCT7 and *SLC16A2*/MCT8 (76% and 71% interaction, respectively) (Appendix A). These trends are in agreement with our findings in the primary dataset, which showed that all four of these SLC16 genes were among the most highly connected MCTs. The strength of MCT–ECM interactions was significantly higher than for ECMs interacting with non-MCT genes (*p*-value = 0.008, Wilcoxon rank-sum test).

In addition to the findings of the hypernetwork modelling, the direct path analysis revealed four significant interactions between the MCTs and ECMs from the Birnbaum dataset that were found in the primary analysis, including the interaction of *SLC16A10*/MCT10 with *COL4A4*, *HYAL1*, and *ANXA10* (*p*-values <0.01) (Appendix A). 

Finally, the RF model performed better in predicting the tissue phenotypes from both the test (30%) and Birnbaum datasets when trained by the MCT–ECM gene cluster refined from hypernetwork modelling than with a random selection of non-differentially expressed genes from the primary dataset (Appendix A).

### 3.4. Lactate and Thyroid Hormone Transporters Correlate with ECMs Involved in Cancer Associated Signalling Pathways

Functional annotation of MCT (*n* = 7) and ECM (*n* = 254, differentially expressed and correlated with MCTs) transcripts using DAVID revealed the presence of ECM components involved in ECM–receptor interactions, protein digestion and absorption, the phosphoinositide-3-kinase–protein kinase B (PI3K-Akt) signalling pathway, the mitogen-activated protein kinase signalling pathway, and pathways in cancer (Appendix A). Extracellular matrix organisation and disassembly, cell adhesion, collagen catabolic process, cell adhesion by integrin, and plasma membrane lactate transport were found as biological processes associated with ECMs and MCTs, respectively. In addition to lactate transport as a biological process for most MCTs, thyroid hormone transport was highlighted for *SLC16A2*/MCT8 and *SLC16A10*/MCT10.

### 3.5. MCT and ECM mRNA Levels Are Not Associated with Age at PDAC Diagnosis

We investigated if the levels of MCT, ECM, and ECM-related mRNAs in stromal and epithelial samples (*n* = 187) associated with age at PDAC diagnosis by a Pearson correlation analysis. The age at diagnosis ranged between 38 and 89, with a mean and median of 68.3 and 69, respectively.

We found that there was no strong correlation between the levels of any of these mRNA transcripts with age at diagnosis (Appendix A).

### 3.6. SLC16A3/MCT4 and Several ECM Components Are Significantly Upregulated in STS Subjects

When we examined the expression of the top four MCT genes, including *SLC16A5*/MCT6, *SLC16A7*/MCT2, *SLC16A10*/MCT10, and *SLC16A1-AS1*, in stromal–epithelial samples in short- and long-term survivors of PDAC (14 and 13 subjects, respectively) from the GSE79668 dataset, we found that none of these MCT genes showed differential expression in these subjects (FDR > 0.05). The only MCT gene that was significantly upregulated in STS subjects was *SLC16A3*/MCT4 (FDR = 0.02, logFC = 1.55). However, the unadjusted *p*-values of these MCTs indicated a significant upregulation of *SLC16A7* (logFC = 1.3) and downregulation of *SLC16A1-AS1* (logFC = −1.0) in STS subjects (*p*-values of 0.003 and 0.006, respectively) (Figure 4).

Among the ECM and ECM-related genes that were differentially expressed in the stromal–epithelial samples (*n* = 255), only 15 of them were significantly expressed, with the majority being upregulated in the STS subjects (Figure 5, FDR < 0.05), including *CTSH*, *S100A2, SERPINE1*, *WNT7B, GPC3*, and a number integrins (*ITGA3* and *ITGA6*).

## 4. Discussion

PDAC is a devastating cancer with high morbidity and low survival rates worldwide, with no effective medical treatments available to improve patient prognosis. The drawbacks of the existing treatment strategies for PDAC and lack of improvement in patient outcomes necessitate the need for a better understanding of the tumour microenvironment and the mechanisms contributing to disease progression [5,7,46]. In this study, we hypothesised that MCTs contribute to PDAC desmoplasia by associating with ECM components. We sought to explore the presence of a causal relationship between MCTs and ECMs to suggest that the increased expression of ECM components may be linked with increased MCT expression.

Our first aim was to understand the expression levels of MCT mRNAs in PDAC biopsies relative to non-tumour tissues as such a comparison has not been widely explored using RNA-seq data. The unsupervised segregation and hierarchical clustering analysis of the collated data from GEO showed no separation of the tumour and non-tumour samples, indicating the similarity of these tissue types. In addition, the hierarchical clustering of tumour and non-tumour samples indicated large interstudy variations in gene expression. The large intertumoral variations may perhaps be explained by the highly heterogenic nature of PDAC disease and, consequently, the biopsy samples that were also highlighted in previous studies [4,35]. In addition to biological variations, technical factors such as the procedures used in tissue acquisition, the sequencing platform, and the library preparation method may also contribute to such heterogeneity. Even with the use of the same sample acquisition technique, differences in the other mentioned factors may still lead to variations in the data. For example, close examination of the tumour samples in Figure A3a,b shows some separation of the tumour samples from the GSE93326 and GSE79668 datasets, although both studies employed macrodissection of the bulk tissue following surgical resection [35,48]. It was also interesting to see the lack of separation between the tumour and non-tumour samples, which may be influenced by the purity of the bulk tissue during sample collection. Another factor that might have played a role could be the lack of properly paired samples, as we only had a single dataset, which included paired tumour and non-tumour tissues with limited sample sizes with the remaining datasets, not including non-tumour tissues. As a result of the large patient variability and the limited sizes of non-tumour samples, we concluded that conducting a differential gene expression analysis on tumour and non-tumour samples using the combined dataset would be inappropriate.

We subsequently focused on the stromal and epithelial dataset. This is because MCTs are reported in both the stroma and epithelium, and in PDAC, the main ECM production happens in stromal cells. Hence, it is important to understand and correlate the expression of MCTs and ECMs in different compartments. Our analysis revealed nine differentially expressed MCT genes in both stromal and epithelial samples, including the upregulated thyroid hormone (TH) transporters MCT8 (*SLC16A12*) and MCT10 (*SLC16A10*), monocarboxylate transporter MCT2 (*SLC16A7*), MCT6 (*SLC16A5*) with a potential role in glucose and lipid metabolism, and an lncRNA *SLC16A1-AS1* previously identified as a prognostic or diagnostic biomarker in a number of cancers [20,54,55,56,57,58,59,60,61]. To our knowledge, the upregulation of the TH transporters in the stromal–epithelial samples has not been highlighted previously. TH transporters regulate the availability of THs T_3_ and T_4_ in cells based on their levels in the local tissue. THs themselves are key regulators of energy metabolism, growth, differentiation, and the physiological function of tissues [54,62]. The role of THs in cancer has been pointed in several studies and their association been summarised in a recent review [63]. For example, the induction of cell proliferation and metabolism by T_3_ via activation of the TRβ1/Akt pathway was demonstrated in human pancreatic insulinoma cells [64]. In addition, the inhibition of TH binding to their cell surface receptor integrin αVβ3 was shown to inhibit tumour cell proliferation and angiogenesis in in vitro and in vivo xenograft models of pancreatic cancer [65]. A more recent study proposed pharmacologically induced TH inactivation as a strategy to reduce tumour metastatic potential based on their observation that T_3_ promoted epithelial–mesenchymal transition by the transcriptional activation of ZEB-1, mesenchymal genes, and MMPs and suppression of E-cadherin, thereby influencing tumour progression and metastasis in skin squamous cell carcinoma [66]. Furthermore, a link between THs and ECM organisation has been shown, where T_3_ was suggested to induce the secretion of growth factors that stimulate cellular proliferation, ECM reorganisation involving fibronectin and laminin, and changes in cell spreading and adhesion [67]. In contrast to the influence of THs on tumour progression discussed above, the synergistic action of T_3_ in combination with gemcitabine and cisplatin was shown to enhance the cytotoxicity of the chemotherapy agents in in vitro models of pancreatic cancer, highlighting the benefits of combination therapy for the treatment of pancreatic cancer [68]. Despite the presence of associations between thyroid dysfunction, including hypothyroidism and the increased risk of pancreatic cancer, the effects of THs in cancer neoplasia is currently conflicting [47,56,62]. Therefore, a better understanding of the molecular mechanisms involved in the contribution of THs to pancreatic cancer is needed. The lncRNA *SLC16A1-AS1* has previously been found to be upregulated in osteosarcoma, glioblastoma, and oral squamous cell carcinoma [50,52,63]; downregulated in non-small cell lung cancer and cervical squamous cell carcinoma [26,56]; and show conflicting expression profiles in hepatocellular carcinoma [51,53,54]. This is the first study highlighting the differential expression of *SLC16A1-AS1* in PDAC, showing lower levels of expression in the epithelial relative to the stromal compartment. With increasing number of reports showing *SLC16A1-AS1* as a potential biomarker in cancer, we propose an investigation to understand how the upregulation of this lncRNA in PDAC associates with tumour development, progression, and overall survival.

In addition to the MCT genes mentioned above, we found several mitogenic growth factors and their receptors such as *EGFR* (epidermal growth factor receptor), *IGF1* (insulin-like growth factor 1), *IGF1R* (insulin-like growth factor 1 receptor), *FGF1/7* (fibroblast growth factor 1/7), *FGFR1/2* (fibroblast growth factor receptor 1), and *VEGFB/C* (vascular endothelial growth factor B/C) to have significant differential expressions in the stromal–epithelial dataset. In the majority of cases, these signalling molecules showed upregulation in the stromal compartment, presumably secreted by the PSCs of the tumour microenvironment, contributing to angiogenesis and tumorigenesis [69,70,71]. Epithelial-to-mesenchymal transition (EMT), which has a crucial role in fibrosis and the formation of metastasis, is associated with a poor prognosis [4]. EMT during PDAC progression is induced by both hyaluronic acid and collagen (non-soluble) and soluble components of the ECM [7]. In this study, the analysis of the stromal–epithelial dataset showed significant upregulation of the relevant genes, such as HAS2 (hyaluronan synthase 2), *COL1A1/2* (collagen-type I alpha 1/2 chain), *FGF1/7*, HGF (hepatocyte growth factor), *NOTCH1/3/4* (Notch receptor 1/3/4), *TGFB1/3* (transforming growth factor beta 1/3), and *WNT2/2B/4/5A* (Wnt family member 2/2B/4/5A) in the stromal compartment, indicating their potential contribution to the EMT in PDAC. Our second primary goal was to explore the causal relationship between MCTs and ECMs in PDAC and understand if the expression of the latter is influenced by the upregulation of MCTs. We used hypernetworks to investigate the relationships between MCTs and ECMs and tease out the functional connectivity among the individual elements. Hypernetworks summarise the conditional independence between genes in a network model by using all the interaction data, not just pairwise [28]. This defines a roadmap of the relationships between genes where the “width” of the roadways is recognised as defining the importance of the higher-order organisation that represents the generation of the mechanism [72]. These features of hypernetworks are considered to support a causal model [73], although we recognise that the validation of bidirectional interactions would require further supporting laboratory work. Hypernetworks, therefore, highlight causal links between biological pathways. Recently, the modelling of conditional transcriptomic interactions has been shown to improve prognostic predictions in cancer [74], and we have used random forest to highlight the potential predictive value of our findings derived from the hypernetwork. Our analysis indicated seven MCTs with functional association in a pairwise manner, including the lncRNA *SLC16A1-AS1*, MCT7 (*SLC16A6*), MCT6 (*SLC16A5*), MCT10 (*SLC16A10*), MCT8 (*SLC16A2*), MCT13 (*SLC16A13*), and MCT2 (*SLC16A7*). The extraction of the genes associated with MCTs from the hypernetwork matrix revealed 255 ECMs that showed a correlation with 90% of these MCTs, including several collagens, laminins, fibronectin 1, and the ECM remodelling enzymes. A similar analysis conducted for the ECMs indicated 376 ECM transcripts that were functionally connective and correlated with four MCTs, including the differentially expressed *SLC16A1-AS1*, *SLC16A2*/MCT8, and *SLC16A10*/MCT10. Hypernetworks generated for both MCTs and ECMs demonstrate the presence of bidirectional associations and suggest that a causal link exists between these two sets of transcripts. To rule out the randomness of this observation, we conducted a robustness testing by firstly generating hypernetworks with random transcripts that were of identical sizes to the number of MCTs and ECMs that were clustered in our analysis. The difficulty we experienced in executing iterations with random transcripts in replacement of MCTs showed that the relationships between MCTs and ECMs were stronger than those that were frequent between random transcripts (Table 3). When the analysis was repeated for ECMs, we found 2.45 ± 1.58 MCTs that were associated with ECMs, similar to our primary findings. As a result, we further conducted a direct path analysis that aimed to refine the correlation matrix of MCTs and ECMs and revealed only the direct gene–gene interactions with a causal relationship by silencing indirect connections. Correlations in biological networks that retain both direct and indirect links can confound the identification of direct interactions [40]. Independent from the hypernetwork modelling approach used, the direct path analysis confirmed the presence of significant MCT–ECM interactions, involving particularly *SLC16A10*/MCT10, which were among the strongest of all interactions observed within the whole silenced network. The bidirectional relationship between the MCTs and ECMs and the presence of significant direct interactions, particularly with *SLC16A10*/MCT, were both evident from the hypernetwork and direct path analysis conducted using the Birnbaum dataset, directly validating our primary findings. Despite the high intersample variation in PDAC tissues and differences in the gene expression levels often seen between studies, our ability to validate the primary findings with an independent dataset is paramount. This analysis therefore further reinforces that the connectivity of MCTs and ECMs is not by random chance, and the causal relationship between the MCTs and ECM action is robust. The mechanisms of the direct interactions between the MCTs and ECMs identified in this study are currently unknown and requires further elucidation. One potential facilitator of this interaction may be CD147, an MCT chaperone and regulator, which is a key contributor to tumour growth and metastasis by promoting ECM remodelling through the induction of MMPs [75]. The expression of CD147 is influenced by its association with MCTs, and the synergistic action of both proteins could enhance metastasis via acidification of the tumour microenvironment and degradation of ECM by MMPs [24].

The tumour biomarkers are undoubtedly valuable in several aspects of patient management, including screening, diagnosis, monitoring, or patient stratification for an associated cancer therapy. Given the insidious nature and poor prognosis of PDAC, the identification of novel biomarkers for early detection, personalised therapy, and post-resection follow-up are urgently needed to improve patient overall survival. As ECM is a driver of tumorigenesis, the use of ECM-derived biomarkers can immensely facilitate the diagnosis and prognosis of patients with cancer in the clinic [76]. To this end, we evaluated the presence of correlations between the levels of the MCT and ECM components in stromal and epithelial samples and age at PDAC diagnosis. Many of the ECM components that showed differential expression and were highlighted in the hypernetwork analysis, including *MMPs*, *TIMP3*, *FN1*, *LAMC1*, and *COL4A1/6A2*, have previously been reported as prognostic or diagnostic markers in different cancers [76]. In our analysis, we found no association between any of the MCTs or ECM components with age at diagnosis. However, scrutinising the expression levels of these ECM components in short-term survivors of PDAC revealed a number of genes that were upregulated in these subjects and have been suggested to have a diagnostic, predictive, or prognostic value in different cancers, including *S100A2* (pancreatic cancer), cathepsin H/*CTSH* (thyroid carcinoma), *SERPINE1* (gastric cancer and oesophageal cancer), *WNT7B* (breast cancer and colorectal cancer), and *GPC3* (hepatocellular carcinoma) [77,78,79,80,81,82,83,84]. Although the latter analysis using the GSE79668 dataset comes from a rather small dataset and poses a limitation, the findings still highlight some potential biomarkers that could be investigated in PDAC. Finally, of the significant interactions identified in the direct path analysis, *SLC16A10*/MCT10 showed the strongest interactions with *HYAL1* and *ANXA10*, genes proposed to have a prognostic value in patients with PDAC [85,86], which opens an avenue for further investigation.

## 5. Conclusions

Our transcriptomic analysis revealed multiple MCTs in PDAC stromal and epithelial compartments, including lncRNA *SLC16A1-AS1*, which may serve as a novel potential diagnostic or prognostic biomarker. In addition, a subset of the differentially expressed ECM components showed associations with MCTs, such as collagens, laminins, fibronectin 1, and ECM crosslinking and remodelling enzymes, highlighting the role of MCTs in PDAC desmoplasia, which should be considered when developing future treatment strategies to improve patient outcomes. Analysis of the higher-order interactions through hypernetworks indicates the presence of a causal link between MCTs and ECMs and warrants the need for further studies to elucidate their functional connections.

## Figures and Tables

**Figure 1 cancers-14-01298-f001:**
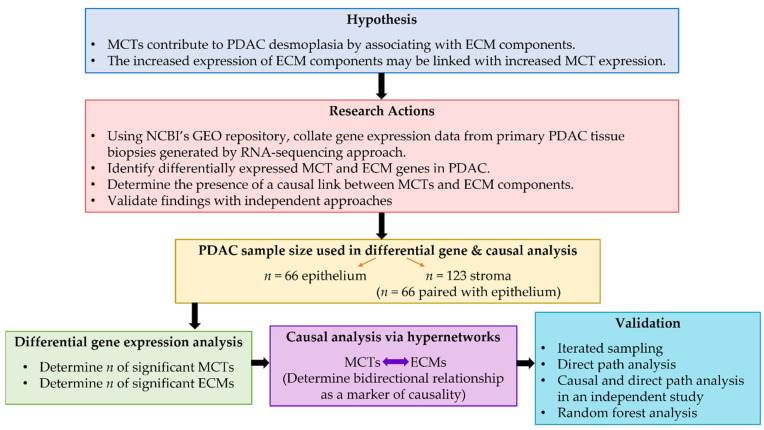
Study hypothesis and actions taken to conduct a causal analysis to determine the link between MCTs and ECMs using epithelial and stromal data from patients with PDAC. A differential gene expression analysis was first conducted to determine significant MCT and ECM genes to refine the causal analysis performed using a hypernetwork model. The findings were validated using four independent approaches.

**Figure 2 cancers-14-01298-f002:**
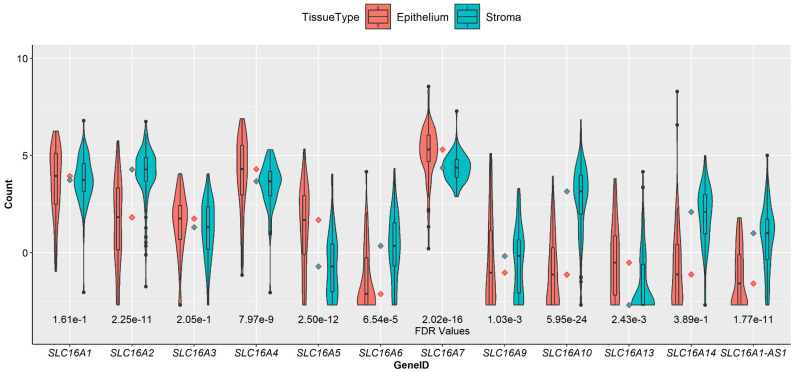
Violin plots of log-normalised expression values of MCT genes in PDAC stroma (*n* = 123) and the epithelium (*n* = 66). FDR indicates a false discovery rate of 5%. Four MCT genes showed significant upregulation in the stroma (*SLC16A2/6/10* and *SLC16A1-AS1*), with 5 MCTs being significantly upregulated in the epithelium (*SLC4/5/7/9/13*) and 3 showing no difference in expression between the two tissues. *SLC16A4* and *SLC16A10* showed the largest upregulation in the epithelium and stroma, respectively.

**Figure 3 cancers-14-01298-f003:**
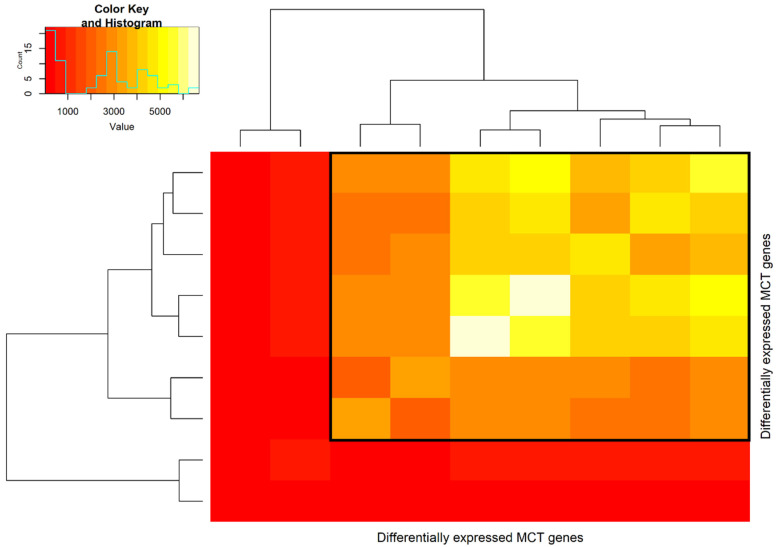
Hypernetwork heatmap of MCT transcripts (*n* = 9) in PDAC stroma and epithelium. Colour intensity represents the number of binary relationships with the rest of the transcriptome (*n* = 13,815 total transcripts) shared between a pair of MCT transcripts. Two clusters were identified, with cluster 1 having seven MCTs, including *SLC16A1-AS1*, *SLC16A6*, *SLC16A5*, *SLC16A10*, *SLC16A2*, *SLC16A13*, and *SLC16A7* (highlighted in black box), and cluster 2 having two MCTs, including *SLC16A9* and *SLC16A4*.

**Figure 4 cancers-14-01298-f004:**
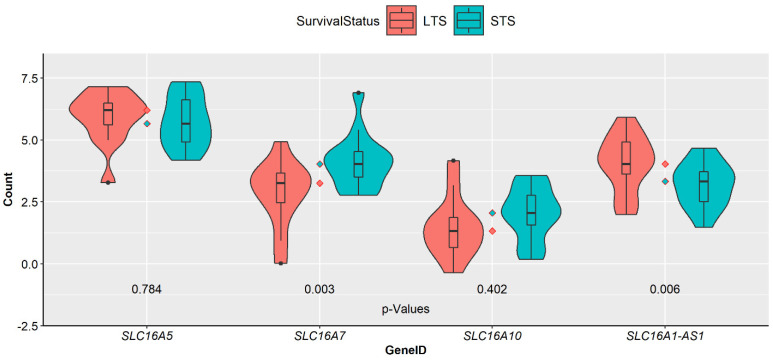
Log-normalised levels of MCT transcripts *SLC16A5*/MCT6, *SLC16A7*/MCT2, *SLC16A10*/MCT10, and *SLC16A1-AS1* in short- and long-term survivors (STS and LTS, respectively) from the GSE79668 dataset. These four genes are the top 4 differentially expressed MCTs in PDAC stroma–epithelium (FDR < 0.05) from the GSE93326 dataset, hence the rationale for selecting them for investigation in the GSE79668 dataset. The number of samples from STS and LTS subjects are 14 and 13, respectively. The indicated *p*-values are not FDR-adjusted. Of the four MCTs investigated, *SLC16A7* and *SLC16A1-AS1* were significantly upregulated in STS and LTS subjects, respectively, with the other two MCTs showing similar expressions in both survival groups (*p* < 0.05).

**Figure 5 cancers-14-01298-f005:**
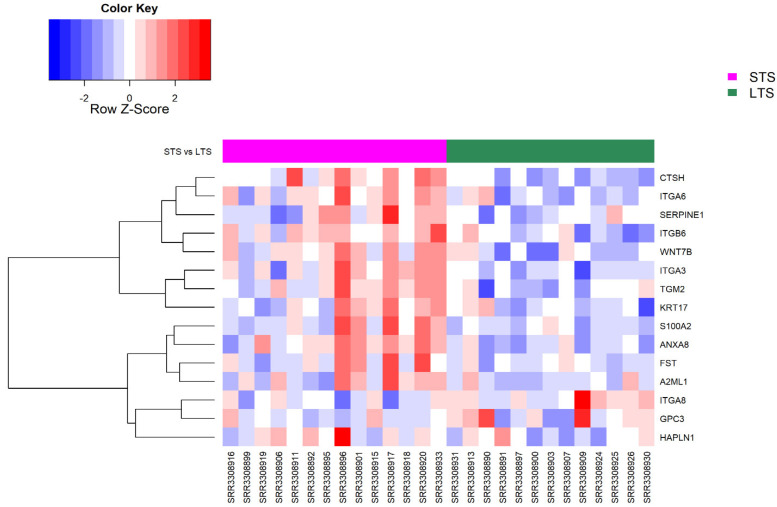
Heatmap and dendrogram showing the normalised expression levels of ECM-related genes (*n* = 15) that were differentially expressed in PDAC stroma–epithelium, as well as in short- and long-term PDAC survivors (STS and LTS, respectively), from the dataset GSE79668 (FDR < 0.05). The number of samples from STS and LTS subjects are 14 and 13, respectively. Red and blue colours indicate increased and decreased expression, respectively.

**Table 1 cancers-14-01298-t001:** Expression of MCT genes in PDAC stroma (*n* = 123) relative to the epithelium (*n* = 66). In total, 12 MCTs were identified, of which 9 showed significant differential expression between the stroma and epithelium, with the remaining MCTs having similar expression in these tissues. In addition to the protein-coding MCT genes, lncRNA *SLC16A1-AS1* was also found to be downregulated in the tumour epithelium.

Gene Name	Protein Name	FDR	LogFC ^‡^	logCPM	*p*-Value
*SLC16A10* ^†^	MCT10	6 × 10^−24^	3.52	3.12	5 × 10^−25^
*SLC16A7* ^†^	MCT2	2 × 10^−16^	−1.17	5.06	3 × 10^−17^
*SLC16A5* ^†^	MCT6	3 × 10^−12^	−2.18	1.54	5 × 10^−13^
*SLC16A1-AS1* ^†^	-	2 × 10^−11^	2.25	1.13	× 10^−12^
*SLC16A2* ^†^	MCT8	2 × 10^−11^	1.63	4.13	5 × 10^−12^
*SLC16A4* ^†^	MCT5	8 × 10^−9^	−1.05	4.29	2 × 10^−9^
*SLC16A6* ^†^	MCT7	7 × 10^−5^	1.48	0.88	3 × 10^−5^
*SLC16A9* ^†^	MCT9	1 × 10^−3^	−1.29	1.09	5 × 10^−4^
*SLC16A13* ^†^	MCT13	2 × 10^−3^	−1.46	0.24	1 × 10^−3^
*SLC16A1*	MCT1	2 × 10^−1^	−0.28	4.25	1 × 10^−1^
*SLC16A3*	MCT4	2 × 10^−1^	−0.30	1.92	2 × 10^−1^
*SLC16A14*	MCT14	4 × 10^−1^	−0.33	2.64	3 × 10^−1^

FDR, false discovery rate; logCPM, log2 count per million gene expression; logFC, log2-fold change between the stroma and epithelium. ^†^ Genes with FDR < 0.05. ^‡^ Positive and negative values indicate upregulation and downregulation in the stroma, respectively.

**Table 2 cancers-14-01298-t002:** The number of MCT and ECM-related transcripts that showed association with ECM and MCT transcripts in the hypernetwork analysis, respectively.

	MCTs (*N* = 7)	ECMs (*N* = 376)
Transcripts correlating with 90% of differentially expressed MCTs/ECMs	ECM-related transcripts (*n* = 255)	*SLC16A2, SLC16A10, SLC16A14, SLC16A1-AS1*

*N* indicates the number of MCT or ECM transcripts in the hypernetwork clusters.

**Table 3 cancers-14-01298-t003:** Assessment of the robustness of the causal relationship between MCT and ECMs by random sampling of transcripts 1000 times. The table shows the number of MCT and ECM-related transcripts that showed correlation with randomly sampled transcripts given the maximum number of iterations achieved.

	Random Transcripts Representative of MCTs (*n* = 7)	Random Transcripts Representative of ECMs (*n* = 376)
Maximum number of iterations achieved	23	1000
Transcripts correlating with 90% of the random transcripts (mean ± SD)	n/a	2.45 ± 1.58

n/a, not applicable.

**Table 4 cancers-14-01298-t004:** The strongest interactions found between differentially expressed MCT and ECM genes within the silenced network. Values from the matrix *S* describing the directness of the relationship between node pairs are presented alongside a z-score describing the position of this value in the whole network and the associated *p*-value of this position. All interactions presented below are statistically significant.

Interacting Genes	Relationship Directness	Z-Score	*p*-Value
“SLC16A10” interacts with “HYAL1”	40.9	3.88	1.29 × 10^−4^
“SLC16A10” interacts with “ANXA10”	35.2	3.35	6.71 × 10^−4^
“SLC16A10” interacts with “MUC5AC”	34.4	3.26	1.33 × 10^−3^
“SLC16A10” interacts with “LGALS4”	32.1	3.04	2.78 × 10^−3^
“SLC16A10” interacts with “CTSE”	31.1	2.95	3.80 × 10^−3^
“SLC16A10” interacts with “PAMR1”	30.4	2.88	4.70 × 10^−3^
“SLC16A10” interacts with “BMP1”	30.2	2.86	5.01 × 10^−3^
“SLC16A10” interacts with “ANGPTL1”	28.6	2.72	5.53 × 10^−3^
“SLC16A10” interacts with “TNFSF12”	28.5	2.71	5.72 × 10^−3^
“SLC16A10” interacts with “PLXNC1”	28.2	2.68	6.29 × 10^−3^
“SLC16A10” interacts with “PI3”	27.3	2.59	8.12 × 10^−3^
“SLC16A10” interacts with “MFAP3”	28.3	2.68	8.53 × 10^−3^
“SLC16A2” interacts with “SEMA3B”	26.8	2.55	9.19 × 10^−3^
“SLC16A10” interacts with “FGF10”	26.8	2.55	9.37 × 10^−3^
“SLC16A10” interacts with “GPC1”	26.7	2.54	9.60 × 10^−3^

The silenced network was visualised in Cytoscape v3.9.0 using aMatReader v1.2.0. The modulus of the silenced matrix score (relationship directness) are shown.

## Data Availability

Publicly available datasets were analysed in this study. The data can be found in the GEO repository: GSE119794, GSE93326, GSE79668, GSE131050, and GSE164665. The source codes are available at the GitHub repository: https://github.com/aysheu/pancreatic-cancer-pipeline, accessed on 25 February 2022.

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
