# Peer review of "Monocarboxylate Transporters Are Involved in Extracellular Matrix Remodelling in Pancreatic Ductal Adenocarcinoma"

_cancers, 2022, doi:10.3390/cancers14051298_

Round 1

Reviewer 1 Report

In terms of methodology and technical aspects of the paper, I have a few concerns:

  • The authors state that they use PCA for clustering analysis. However, PCA is not a clustering algorithm, but a simple linear dimensionality reduction technique. Not only is it not clear what the authors are doing, but this whole part raises questions as to the authors' understanding of the methodologies used to analyse the data, all of which badly affects the credibility of the work and its results.

  • The authors claim that they examine causality; yet, all they do is establish pair-wise correlations.

There are also issues with presentation:

  • Keywords should not include terms already in the title of the paper. The very point of keywords is to provide additional indexing information. Yet, 2 of the 3 keywords are in the title.

  • In "pca function" the authors should capitalize "PCA" or, if they are referring to a specific function in the package, use a fixed width font to communicate this.

  • The p-values reported (e.g. in Table 1) should not include so many significant digits. It should be clear that all but the leading digit are simple stochastic effects (relate this to my first comment above).

Author Response

1) In terms of methodology and technical aspects of the paper, I have a few concerns:

a) The authors state that they use PCA for clustering analysis. However, PCA is not a clustering algorithm, but a simple linear dimensionality reduction technique. Not only is it not clear what the authors are doing, but this whole part raises questions as to the authors' understanding of the methodologies used to analyse the data, all of which badly affects the credibility of the work and its results.

Authors: We thank the Reviewer for this comment. We used PCA to segregate the samples based on their similarities and differences and enable interpretation of the data. We appreciate the use of the term “clustering” would be misleading, therefore, to address the Reviewer’s comment, we have now revised the terminology and changed this to ‘segregation’ instead in lines 150, 163, 324, 334, 337, 559, 795, 803, and 804.

b) The authors claim that they examine causality; yet, all they do is establish pair-wise correlations.

Authors: We appreciate the Reviewer’s concern. However, our analysis does not represent simple pair-wise correlations. In order to investigate causality, we used hypernetworks which describe complex or high-dimensional relationships between transcripts by flattening them into a single metric of similarity via summarizing the correlation matrices. We have now emphasized this point at the end of our introduction in line 119. To bring clarity on this matter further and avoid similar concerns by other readers, we have now described this process more clearly in the discussion (lines 643-654). We have also clarified the relationship of the hypernetworks to causality in the discussion and added further supporting literature (lines 645, 648, and 652). We have represented the direct path analysis using a more robust interpretation of the silencing matrix, this refines stronger direct interactions from our causal analysis (Table 4, line 455).

2) There are also issues with presentation:

a) Keywords should not include terms already in the title of the paper. The very point of keywords is to provide additional indexing information. Yet, 2 of the 3 keywords are in the title.

Authors: We acknowledge the Reviewers’ comment on the keywords. We have now revised all keywords with those that have not been used in the title but are relevant for the subject of our study in line 45.

b) In "pca function" the authors should capitalize "PCA" or, if they are referring to a specific function in the package, use a fixed width font to communicate this.

Authors: We thank the Reviewer for this comment. We are referring to a specific function, therefore we have now revised the font of the word “pca” in the manuscript to a fixed width font in line 160.

c) The p-values reported (e.g. in Table 1) should not include so many significant digits. It should be clear that all but the leading digit are simple stochastic effects (relate this to my first comment)

Authors: We appreciate the Reviewer’s comment and have now reduced the number of significant digits for both unadjusted and FDR-adjusted p-values in Table 1 (line 361) of the manuscript.

Reviewer 2 Report

The present manuscript describes the correlation between the role of MCTs in desmoplasia through their associations with ECM components in PDAC.

The in silicon data is very interesting but some wet data would have been of interest to confirm the correlation described by the authors.

Author Response

Reviewer 2: The present manuscript describes the correlation between the role of MCTs in desmoplasia through their associations with ECM components in PDAC. The in silico data is very interesting but some wet data would have been of interest to confirm the correlation described by the authors.

Authors: We appreciate the Reviewer’s comment and agree that experimental data would be beneficial to confirm our findings. To reflect this, we have already emphasized the necessity of further studies in our discussion and conclusion sections in lines 649 and 729, respectively. At this stage, we do not have the necessary funding to conduct these studies. However, our in silico modelling has been conducted using multiple approaches and validated in an independent dataset, therefore constitutes a strong finding that should be reported before future bench work is conducted.

Reviewer 3 Report

Ayşe Ufuk et al. uncovered  Monocarboxylate Transporters to be potentially involved in Extracellular Matrix Remodelling in Pancreatic Ductal Adenocarcinoma - PDAC.

Points to consider:

  1. An extra validation would be required
  2. Sample size is critical: an in silicon analysis should always include an original validation dataset/datalake
  3. The exact nature of the study is not perfectly clear: I would suggest to restructure as follows: 

    The elements of a PICOT question are:

    P (Patient, population or problem)

    Who or what is the patient, population or problem in question?

    I (Intervention)

    What is the intervention (action or treatment) being considered?

    C (Comparison or control)

    What other interventions should be considered?

    O (Outcome or objective)

    What is the desired or expected outcome or objective?

    T (Time frame)

    How long will it take to reach the desired outcome and what treatment are envisioned

  4. Study limitations (see points 1-3) should be highlighted.
  5. This reviewer personally misses some insights regarding background and discussion about PDAC signaling pathways, genetic alterations and their relevance for tumor microenvironment: please refer to PMID: 33918146 and expand the introduction/discussion sections.

Author Response

Reviewer 3:

Points to consider:

  1. An extra validation would be required

Authors: We appreciate the Reviewer’s comment. To address this matter, we have now included additional validation steps including our use of an independent dataset to validate the causal hypernetwork and direct path analyses, as well as the random forest model to evaluate the ability of the training set from our primary dataset to correctly classify the tissue phenotypes in the validation dataset. The methods and results sections related to validation can be found in lines 252-280 and 465-494, respectively.

2. Sample size is critical: an in silicon analysis should always include an original validation dataset/datalake

Authors: We appreciate the Reviewer’s comment. We now present a full validation of our findings in an independent dataset. As we had no expectation of an effect size in our training data, the provision of further validation is sufficient to support the analysis presented. As we have now established these interactions and the effect size, future work can use appropriate modelling to identify the critical sample size.

3. The exact nature of the study is not perfectly clear: I would suggest to restructure as follows: 

The elements of a PICOT question are:

P (Patient, population or problem)

Who or what is the patient, population or problem in question?

I (Intervention)

What is the intervention (action or treatment) being considered?

C (Comparison or control)

What other interventions should be considered?

O (Outcome or objective)

What is the desired or expected outcome or objective?

T (Time frame)

How long will it take to reach the desired outcome and what treatment are envisioned

Authors: We appreciate the Reviewer’s comment. As our research study is not clinical in nature, it is challenging to describe the nature of our work using the PICOT format. However, to address the Reviewer’s request, we have added a new figure in the manuscript (Figure 1) to summarise our research hypothesis and actions we have undertaken to conduct the causal analysis. Please find this in line 118.

4. Study limitations (see points 1-3) should be highlighted.

Authors: We appreciate the Reviewer’s comment. The large inter-study variation we found was a big limitation of the study which prevented us from utilizing additional datasets from the PDAC primary tumours. This was highlighted in lines 577-580 of the manuscript. We also described the limitations of our causal analysis in the discussion in lines 648-650.

5. This reviewer personally misses some insights regarding background and discussion about PDAC signaling pathways, genetic alterations and their relevance for tumor microenvironment: please refer to PMID: 33918146 and expand the introduction/discussion sections.

Authors: We thank the Reviewer for this comment. Using the reference paper the Reviewer has recommended, we have now expanded both the introduction and discussion sections to provide some background and findings in the context of PDAC genetic alterations, signalling pathways, and their relevance for tumour microenvironment. Please find these sections in lines 55-64 and 624-639.

Round 2

Reviewer 1 Report

I am happy with how the authors have addressed my concerns.

Reviewer 3 Report

The authors have clarified several of the questions I raised in my previous review. Most of the major problems have been addressed by this revision.